# Timeline of FDA-Approved Targeted Therapy for Cholangiocarcinoma

**DOI:** 10.3390/cancers14112641

**Published:** 2022-05-26

**Authors:** Su Min Cho, Abdullah Esmail, Ali Raza, Sunil Dacha, Maen Abdelrahim

**Affiliations:** 1Department of Medicine, Houston Methodist Hospital, Houston, TX 77030, USA; scho@houstonmethodist.org; 2Department of Medicine, Weill Cornell Medical College, New York, NY 10065, USA; 3Department of Medicine, Texas A&M College of Medicine, Bryan, TX 77807, USA; 4Section of GI Oncology, Department of Medical Oncology, Houston Methodist Cancer Center, Houston, TX 77030, USA; aesmail@houstonmethodist.org; 5Department of Gastroenterology, Houston Methodist Hospital, Houston, TX 77030, USA; araza@houstonmethodist.org (A.R.); sdacha@houstonmethodist.org (S.D.); 6Cockrell Center of Advanced Therapeutics Phase I Program, Houston Methodist Research Institute, Houston, TX 77030, USA

**Keywords:** cholangiocarcinoma, CCA, cancer, targeted therapy, FDA, biliary

## Abstract

**Simple Summary:**

Cholangiocarcinoma constitutes around 3% of gastrointestinal cancers, and mortality from this cancer has been rising in the recent decades. Many cases of cholangiocarcinoma are unfortunately discovered in their advanced stages which cannot be treated with surgical resection alone. Targeted therapy is a type of medical treatment that has garnered significant interest due to its ability to specifically target cancer cells while sparing normal healthy cells. A few targeted therapies have just recently been approved by the United States FDA for the treatment of cholangiocarcinoma specifically. This manuscript seeks to explore the timeline of targeted therapies with either FDA approval or FDA breakthrough therapy designation. The official approval of these therapies marks a new age for the treatment of cholangiocarcinoma and brings new options for clinicians across the nation for this unfortunate disease.

**Abstract:**

Cholangiocarcinoma (CCA) represents approximately 3% of gastrointestinal malignancies worldwide and constitutes around 10–15% of all primary liver cancers, being only second to hepatocellular carcinoma. Mortality from CCA has been on the rise in recent decades, and in the United States alone there has been a 36% increase in CCA from 1999 to 2014, with over 7000 CCA mortalities since 2013. Targeted therapies, which have been gaining interest due to their greater specificity toward cancer cells, have only recently started gaining FDA approval for the treatment of CCA. In this manuscript, we will go through the timeline of current FDA-approved targeted therapies as well as those that have gained FDA breakthrough therapy designation.

## 1. Introduction

Cholangiocarcinoma (CCA), in essence, is an umbrella term which denotes cancer arising from the biliary tree. Multiple subtypes of cholangiocarcinoma exist depending on its anatomical origin: intrahepatic (iCCA), perihilar (pCCA), and distal extrahepatic (dCCA) [1]. Intrahepatic CCA is located within the hepatic parenchyma and lies in the region proximal to the second-order bile ducts. pCCA lies beyond the second-order bile ducts up to the insertion site of the cystic duct into the common bile duct. dCCA is located in the common bile duct beyond the cystic duct insertion site. Perihilar CCA and distal CCA have previously been collectively classified as extrahepatic cholangiocarcinoma (eCCA) [2]. CCA represents approximately 3% of gastrointestinal malignancies worldwide and constitutes around 10–15% of all primary liver cancers, being only second to hepatocellular carcinoma. CCA presents a major challenge for clinicians because the disease is often found at an advanced stage which limits therapeutic options. Unfortunately, mortality from cholangiocarcinoma has been on the rise in recent decades, primarily driven by the rise in the incidence of iCCA [3]. In contrast, the incidence of pCCA and dCCA have been decreasing [3,4]. The mortality rate of iCCA was approximately 1–2/100,000 and that of eCCA was below 1/100,000 [3]. Just in the United States (US) alone, there has been a 36% increase in CCA mortality from 1999 to 2014, with over 7000 CCA mortality after the year 2013 [5].

To simplify, the etiology for CCA can be broadly divided into two categories: fluke-related and non-fluke-related. In endemic parts of the world such as southeastern and eastern Asia, parasitic liver fluke infections such as with the Opisthorchiidae or Clonorchis families play major roles in the pathogenesis of CCA [6,7]. These infections are usually caused by *Opisthorchis viverrini* and *Clonorchis sinenesis,* which inhabit the biliary tract and thus lead to chronic inflammation [8]. The incidence of CCA can therefore be quite elevated in endemic regions, for example, it can be as high as 100 per 100,000 people in certain regions of Thailand [9]. In other parts of the world where liver flukes are not endemic, some of the risk factors for CCA have included primary sclerosing cholangitis, hepatolithiasis/choledocholithiasis, Caroli’s disease, congenital hepatic fibrosis, choledochal cysts, viral hepatitis B and C infections, liver cirrhosis, chemical exposures, diabetes, and obesity [8,10,11,12,13,14,15,16,17,18]. Almost all of these risk factors are related to chronic inflammation to the biliary tract. In Western countries, primary sclerosing cholangitis has been very well documented as a strong risk factor for CCA, and such individuals can have up to 13% lifetime risk of developing CCA [19].

In recent decades, several modes of treatments have been utilized to fight CCA. Depending on the type of CCA and the locality of the disease, surgical resection remains a treatment option with long-term survival [20]. This includes liver transplantation which has treated hepatobiliary malignancies with significantly improved outcomes [21,22,23,24]. Unfortunately, CCA is often diagnosed at an advanced stage where surgical intervention will not be sufficient. In the case of irresectable, recurrent, or metastatic disease, systemic chemotherapy has historically been an option for CCA [25]. For certain disease presentations such as iCCA, chemotherapy has been used in a more focused manner such as with hepatic arterial infusion pump (HAIP) chemotherapy or trans-arterial chemoembolization (TACE) [26,27]. Radiotherapy is another tool to fight CCA, with modern techniques such as stereotactic body radiation therapy being capable of ablating specific areas of the body while sparing surrounding tissues [27]. Of course, various combinations of aforementioned therapies have also been widely utilized [21,22,28,29,30,31,32,33]. 

In recent years, however, there has been growing interest towards a “cleaner” means of combating cancer. The idea of precision medicine has stemmed from the desire to only fight the cancer cells while sparing normal healthy cells, thereby maintaining the integrity of the body. Immunotherapy, despite not specifically targeting the cancer cells, has emerged as a viable therapy option due to its ability to spare healthy cells; however, options for its utilization in the United States (US) for the treatment of CCA remains currently limited with further research required [30,31,33,34]. 

Targeted therapy has also recently garnered substantial interest as a tool to fight cancer. Just as with immunotherapy, targeted therapy is capable of targeting malignant cells in the body and hence preserving adjacent healthy cells. Monoclonal antibodies, which are one major type of targeted therapy, are able to specifically bind to antigens of cancer cells and thereby exhibit anticancer effects by recruiting host immune functions, interrupting the essential cellular functions of cancer cells, or even delivering cytotoxic compounds specifically to cancer cells [35,36]. Small molecular inhibitors constitute the other major type of targeted therapy, and they exhibit anticancer effects via interfering with the intracellular signaling of tyrosine kinases which are essential for cellular function [35]. The complexity of the cellular system exposes countless potential targets and multiple targets have been identified and extensively studied. Many types of cancers have already begun to utilize targeted therapies as a preferred treatment regimen, and only recently have they also started being recognized as a treatment option for CCA. In recent years, several targeted therapies have recently gained the approval of the US Food and Drug Administration (FDA) for treatment against CCA. In this manuscript, we explore the timeline of the US FDA-approved targeted therapy against CCA which has paved way for the utilization of precision medicine for CCA. 

Of note, we specifically explore drugs that have been approved based on studies primarily consisting of CCA patients. We did not include therapies approved for a wide variety of tumors, as the supportive clinical research did not primarily consist of CCA patients. For example, larotrectinib and entrectinib have both received FDA approval for the treatment of solid tumors (including CCA) with NTRK (neurotrophic tropomyosin receptor kinases) gene fusions, the supportive research had a minimal number of actual CCA patients [37]. 

## 2. Targeted Therapies with FDA Approval

The standard first-line systemic therapy for advanced CCA is a combination of cisplatin + gemcitabine, which was shown by the ABC-02 trial demonstrating a superior median progression-free survival (PFS) (8.0 vs. 5.0 months), disease control rate (DCR) (81.4% vs. 71.8%), and overall survival (11.7 vs. 8.1 months) when compared to gemcitabine alone [38]. Second-line systemic therapies for advanced CCA include gemcitabine + platinum, gemcitabine + fluoropyrimidine, fluorouracil-combinations, and more [39], and these therapies were found to have a much smaller benefit with a median PFS of 3.2 months with no significant difference between the various second-line regimens [40,41]. These treatments, however, do not spare the normal cells and can therefore have substantial side effects. 

### 2.1. Pemigatinib

On 17 April 2020, pemigatinib, under the trade name Pemazyre and developed by the biopharmaceutical company Incyte, became the first targeted therapy approved by the FDA for the treatment of CCA [42]. It received accelerated approval specifically for the treatment of previously treated, unresectable locally advanced or metastatic CCA with fibroblast growth factor receptor (FGFR) 2 fusion or other rearrangements [42]. Pemigatinib is an oral small molecular competitive inhibitor of FGFR 1, 2, and 3, and its efficacy in the treatment of CCA was demonstrated in the FIGHT-202 trial, a multicenter, open-label, single-arm, phase II study [43]. The study consisted of a total of 146 participants with FGF/FGFR alterations previously treated with at least one prior systemic therapy and who were given oral pemigatinib in 21-day cycles of 13.5 mg orally once daily for 14 days followed by 7 days off therapy [43]. A cohort of 107 patients with FGFR2 gene rearrangements/fusions treated with pemigatinib had an overall response rate (ORR) of 35.5% (95% CI (confidence interval), 26.5–45.4%), with 2.8% achieving complete response and 32.7% achieving partial response [43]. Of these 107 patients, 61% had one prior systemic therapy, 27% had two prior systemic therapies, and 12% had more than two prior systemic therapies [43]. Overall, 82% (95% CI, 74–89%) of the patients achieved disease control [43]. The median duration of response (DOR) among the responders was 7.5 months (95% CI, 5.7–14.5 months), the median PFS was 6.9 months (95% CI, 6.2–9.6 months), and the median overall survival was 21.1 months (95% CI, 14.8 months to not estimable) [43]. It was interesting to note that patients with other FGF/FGFR alterations failed to achieve a response for the treatment regimen [43] which explains why FDA approval was specifically for CCA with FGFR2 fusion or other rearrangements. The FIGHT-202 trial demonstrated that the therapeutic effects of pemigatinib was favorable when compared to other second-line chemotherapy and targeted therapies, although direct comparisons were limited by differences in the individual study designs, differences in patient populations, and a separate study which showed overall favorable survivability in patients with FGFR alterations compared to those without [43,44]. Hyperphosphatemia was the most common all-grade adverse event (60%), as was similarly reported with other FGFR inhibitors, and it primarily occurred early after treatment initiation [43,45,46]. Other observed AEs included arthralgia, stomatitis, nail toxicities, hyponatremia, abdominal pain, fatigue, pyrexia, cholangitis, and pleural effusion [43]. The significance of pemigatinib cannot be underestimated, as it essentially opened the gateway for additional targeted therapies for the treatment of CCA in the United States. Pemigatinib has subsequently been approved for CCA in other parts of the world, including by the Japanese Ministry of Health, Labour, and Welfare on 23 March 2021 and by the European Commission on 29 March 2021. Figure 1 shows a visual timeline of the targeted therapies with either FDA approval or breakthrough therapy designation for the treatment of CCA, and pemigatinib was the very first to pave way for the others.

### 2.2. Infigratinib

Infigratinib, with the trade name Truseltriq and developed by QED Therapeutics and Helsinn, is a FGFR1-3-specific kinase inhibitor that was FDA-approved for treatment of CCA on 28 May 2021 [47]. It also received accelerated approval for the treatment of patients with prior treated, unresectable locally advanced, or metastatic CCA with FGFR2 or other rearrangements [47]. It became the second targeted therapy to obtain FDA approval for the treatment of CCA. Its efficacy was shown in a multicenter, open-label, phase II study in 2018 consisting of 61 participants with prior systemic therapy who were given oral infigratinib in 21-day cycles of 125 mg once daily for 14 days followed by 7 days off therapy [45]. Among these participants, 32.8% had one prior systemic cancer therapy, 29.5% had two prior systemic therapies, 18% had three prior systemic therapies, and 19.7% had more than three prior systemic therapies [45]. Most of these patients had genetic alterations to their FGFR2 (95.1%), with 48 patients having FGFR2 fusions, 8 patients having FGFR2 mutations, and 3 patients having FGFR amplifications [45]. Overall, the median duration of exposure to infigratinib was 4.7 months, the ORR was 14.8% (95% CI, 7.0 to 26.2%), DCR was 75.4% (95% CI, 62.7–85.5%), and estimated median PFS was 5.8 months (95% CI, 4.3 to 7.6 months) for all 61 patients in the initial trial [45]. Mature results of the same trial in 2021 consisted of 108 patients with FGFR2 fusions or other rearrangements who had previously been treated with a gemcitabine-containing regimen, and after a median follow-up of 10.6 months, the blinded independent central review (BICR)-assessed ORR was 23.1% (95% CI, 15.6–32.2%), with 1 confirmed complete response and 24 partial responses [48]. The most common AEs included hyperphosphatemia (77%), stomatitis (55%), fatigue (40%), and alopecia (38%) [48]. Ocular AEs were also observed, including dry eyes (34%), nail toxicities (18%), and central serous retinopathy-like and retinal pigment epithelial detachment-like events (17%) [48]. Infigratinib has subsequently been approved for CCA by Health Canada on 29 September 2021, and by the Hainan province in China under a special named patient program on 21 December 2021. 

### 2.3. Ivosidenib

On 25 August 2021, Ivosidenib (trade name Tibsovo, developed by Servier Pharmaceuticals), was approved for the treatment of CCA [49]. It became the third targeted therapy to obtain official FDA approval for the treatment of CCA, and as of the time of writing of this manuscript, it is the latest to gain that designation. Ivosidenib is an oral, small-molecule inhibitor of mutant isocitrate dehydrogenase-1 (IDH1), and it was specifically approved by the FDA for adult patients with previously treated, locally advanced, or metastatic CCA with an IDH1 mutation [49]. Its efficacy was demonstrated in the ClarlDHy trial, which was a multicenter, randomized, double-blind, placebo-controlled, clinical phase 3 trial consisting of 185 participants with mutant IDH1 [50]. Patients in the treatment arm orally received 500 mg ivosidenib daily in continuous 28-day cycles [50]. Results published in 2020 showed a significantly improved median PFS compared to that of the placebo (2.7 months vs. 1.4 months) (50). Median overall survival within the intention-to-treat population was 10.8 months for the ivosidenib group (*p* = 0.06) [50]. The 6-month overall survival rate for the ivosidenib group was 67% and the 12-month rate was 58% compared to that of the placebo group with 59% and 38%, respectively [50]. In a follow-up publication in 2021, the study size had slightly expanded to 187 patients, the majority of whom had metastatic disease and had previously undergone at least one prior therapeutic regimen [51]. Approximately 53% of these patients had previously undergone one line of therapy, while approximately 47% had previously undergone two lines of therapy [51]. The final overall survival was 10.3 months (95% CI 7.8–12.4) with ivosidenib and 7.5 months (95% CI, 4.8–11.1 months) with placebo (*p* = 0.09), but when adjusted for crossover, the median overall survival with placebo was 5.1 months (95% CI, 3.8–7.6 months) (*p* < 0.01) [51]. The reported 12-month survival rate was 43% (95% CI, 34–51%) and 36% (95% CI, 24–48%) for the treatment group and placebo, respectively, and the median treatment duration was 2.8 months and 1.6 months, respectively [51]. In addition, patients under the treatment arm had a superior preservation of their quality of life, both physical and emotional, compared to that of the placebo (51). The most common all-grade AE was nausea (ivosidenib 42% vs. placebo 29%), and the most common grade 3 or higher AEs were ascites (ivosidenib 9% vs. placebo 7%), anemia (7% vs. 0%), and hyperbilirubinemia (6% vs. 2%) [51]. Overall, the clinical trial boasted up to a 63% reduction in disease progression or death with ivosidenib in comparison to the placebo among patients with prior chemotherapeutic treatment [50,51].

## 3. Targeted Therapies with FDA Breakthrough Therapy Designation

### 3.1. Zanidatamab

As of the time of writing of this paper, zanidatamab has not yet been formally FDA-approved for the treatment of CCA. However, on 30 November 2020, Zymeworks Inc. announced that zanidatamab, a human epidermal growth factor receptor 2 (HER2)-targeted bispecific antibody, received an FDA breakthrough therapy designation for patients with locally advanced, unresectable, or metastatic HER2-expressing biliary tract cancers, including iCCA, eCCA, or gallbladder cancer (GBC) [52]. The designation was based off the results of a single-arm, open-label, phase 1 clinical trial consisting of 20 participants (5 iCCA, 4 eCCA, and 11 GBC) with confirmed HER2 overexpression who were treated with zanidatamab at a dose of 20mg/kg every 2 weeks [53]. Seventeen patients were evaluable for a response [53]. ORR was 47% (95% CI, 23–72%), disease control rate was 65% (95% CI, 38–86%), and median DOR was 6.6 months (95% CI, 3.2—not estimable) [53]. Treatment-related adverse effects were experienced by 14 out of 20 patients, all of which were grade 1 or 2 in severity and included symptoms such as diarrhea (45%) and infusion-related reactions (30%) [53]. As of the time of writing of this paper, a phase 2 study (HERIZON-BTC-01) of zanidatamab is ongoing. The efficacy of zanidatamab is being studied under all biliary tract cancers so it is not specific to CCA. However, it still presents as a potential agent for treating CCA, specifically in those with advanced HER2-expressing CCA. 

### 3.2. Futibatinib

As with zanidatamab, futibatinib has not yet been formally approved by the FDA for the treatment of CCA. However, on 1 April 2021, Taiho Oncology and Taiho Pharmaceutical announced that futibatinib, an oral highly selective irreversible small-molecule inhibitor of FGFR1-4, was granted a breakthrough therapy designation by the FDA for the treatment of adults previously treated for locally advanced or metastatic CCA with FGFR2 gene rearrangements [54]. The breakthrough therapy designation was based off the FOENIX-CCA2 trial, a multicenter, single-arm phase 2 study consisting of 103 participants—though interim data were primarily based off 67 participants [55]. A majority of these patients (82.1%) had tumors with FGFR2 fusion, and 44.8%, 28.4%, and 26.9% of the patients had previously undergone one, two, and at least three lines of therapy, respectively [55]. Participants were orally given 20 mg futibatinib daily until disease progression or unacceptable toxicity [55]. ORR was 34.3%, all were partial responses, and the disease control rate (DCR) was 76.1% [55]. The median time of response was 1.6 months and the median DOR was 6.2 months (55). Observed adverse effects included hyperphosphatemia (79.1%), diarrhea (37.3%), and dry mouth (32.8%) [55]. 

## 4. Mechanism of Action of Targeted Therapies for CCA

Table 1 briefly lists the mechanism of action as well as the relevant clinical studies for the targeted therapies with current FDA approval or breakthrough therapy designation for CCA. Both pemigatinib and futibatinib are inhibitors against FGFRs. Pemigatinib is a small molecule inhibitor against FGFR1-3, while futibatinib is a small molecule inhibitor against FGFR1-4. FGFRs consist of a group of receptor tyrosine kinases that are responsible for key parts of cellular development [56]. These receptors span across the cellular membrane, containing both the extracellular component capable of binding to ligands and an intracellular component that transmits the tyrosine kinase signaling pathway [57]. The binding of a ligand to the FGFR initiates the receptor dimerization and autophosphorylation of various tyrosine kinases on the intracellular component of the FGFR, subsequently leading to a complex dance of additional protein phosphorylation activating various intracellular pathways [56,57]. The intracellular signal transduction then leads to various vital parts of cellular development including cell proliferation, cell shape, cell migration, cell differentiation, and cell survival [57,58]. It is therefore unsurprising that a dysregulation of this intricate process has been linked to various types of cancer. Alterations of the receptor have been linked to either constitutive receptor activation or aberrant ligand-dependent signaling, ultimately altering downstream signals to drive tumorigenesis [57,58,59]. By inhibiting the receptors, pemigatinib and futibatinib are able to block the downstream signaling pathways and reduce the viability of cancer cells. In 2016, a next-generation sequencing study consisting of 4853 solid tumors revealed that FGFR aberrations were present in 7.1% of tumors [60]. FGFR2 fusions, a specific pattern of genetic alteration, which has been found to be present in 10–15% of iCCA but very minimally in eCCA [61]. With all this in mind, pharmacologic targeting against the FGFR pathway, such as with FGFR inhibitors, has gained much attention with the development of multiple inhibitors and various clinical trials. Pemigatinib and futibatinib are currently the only FGFR inhibitors that have either been FDA approved or currently in FDA breakthrough designation therapy for the treatment of CCA.

Infigratinib is a competitive tyrosine kinase inhibitor specific to FGFR1-3. Receptor tyrosine kinase inhibitors have also revolutionized cancer therapy in recent decades, with now over 40 compounds approved by the FDA for cancer therapy [62]. As previously mentioned, receptor tyrosine kinases are present in the intracellular domain of the overall receptor, and they are activated by receptor dimerization that comes with ligand–receptor binding in the extracellular domain [56,57]. The activation of tyrosine kinases would then lead to additional downstream effects to affect cellular growth. A receptor tyrosine kinase inhibitor essentially blocks this step, and just as with an FGFR inhibitor, it would repress the effects of an aberrant FGFR. 

Ivosidenib is a small-molecular inhibitor of mutant IDH1 which has been shown to be mutated in various cancers such as colorectal cancer, prostate tumors, melanomas, leukemias, and cholangiocarcinoma [63,64]. IDH1 belongs to the isocitrate dehydrogenase family composed of three isozymes: namely IDH1, IDH2, and IDH3, with IDH1 found in the cytosol and peroxisome while IDH2–3 are found in the mitochondria [65]. IDH is a key metabolic enzyme that catalyzes isocitrate into α-ketoglutarate, a process which also generates NADPH and thus preventing oxidative damage via a reduction in glutathione and thioredoxins [66]. However, a mutant IDH1 developed a neomorphic activity which converts α-ketoglutarate into 2-hydroxyglutarate (2-HG) with the usage of NADPH as a cofactor [67]. The accumulation of 2-HG as a result of a mutant IDH leads to histone and DNA hypermethylation which appears to block cellular differentiation via epigenetic modifications, ultimately leading to tumorigenesis and progression [65,67,68,69,70,71,72]. By inhibiting the mutant enzyme, ivosidenib reduces cancer viability by preventing high levels of 2-HG and preventing differentiation blocks. Given the tumorigenic role of mutant IDH, various IDH inhibitors have been developed and tested in clinical trials. IDH1 and IDH2 mutations have been reported in approximately 15–20% of iCCA, with IDH1 mutations more common than IDH2 [73]. Overall, given the prevalence of such mutations in CCA, IDH inhibitors could potentially play a substantial role as a targeted therapy against CCA, particularly iCCA. 

Zanidatamab is a bispecific antibody which targets HER2 and belongs to a family of human epidermal growth factor receptors. An aberrancy of the HER family of receptors that has been well documented is its key role in tumorigenesis, with HER2 amplification or overexpression being implicated in various types of cancer including breast cancer, gastric/gastroesophageal cancer, ovarian cancer, colorectal cancer, lung cancer, bladder cancer, and head/neck cancer [74]. HER functions very similarly to FGFR in that it regulates vital normal cellular processes such as growth, survival, and differentiation. Just as with FGFR, HER has both an extracellular component that binds ligands and an intracellular tyrosine kinase domain which thereafter becomes activated to activate other cellular pathways and downstream effects. An amplification or overexpression of the receptors results in increased responsiveness to growth factors and cellular signaling which ultimately leads to malignant growth [75]. Zanidatamab is able to bind to the two extracellular domains of HER2 which leads to receptor clustering, receptor internalization, and downregulation [76,77], in turn leading to the inhibition of uncontrolled downstream signaling and the prevention of malignant growth. Figure 2 shows a simplified summary of the mechanism of action of targeted therapies with current FDA approval or breakthrough therapy designation.

## 5. Future Steps

The targeted therapies that have gained FDA approval or breakthrough therapy designation for the treatment of CCA to date consist of two FGFR inhibitors, one FGFR-specific tyrosine kinase inhibitor, one IDH inhibitor, and one HER-targeting antibody. These five agents constitute merely a small fraction of the total pharmacologic agents currently under investigation for the treatment of various malignancies including CCA [78]. Not only are numerous drugs being synthesized and studied for the same aforementioned targets, but are also pharmacologic agents being studied for multiple other targets. Additional targets for tyrosine kinase receptors include vascular endothelial growth factor receptor (VEGFR), platelet-derived growth factor receptor (PDGFR), epidermal growth factor receptor (EGFR), hepatocyte growth factor receptor (HGFR), and ROS1 receptor tyrosine kinase [78]. Targets against cytoplasmic proteins in the downstream signaling cascade have also been under investigation, since mutations in these proteins have also been documented in various malignancies [79,80]. These include proteins in the RAS/RAF/MEK/ERK signaling pathways and PI3K/AKT/mTOR signaling pathway [78]. Certain CCAs have BRAF-V600E mutations, as seen in other forms of malignancies such as CRC and melanoma [81]. There has recently been a project headed by the University of Texas MD Anderson Cancer Center that has had promising results by combining dabrafenib, a BRAF inhibitor, and trametinib, a MEK inhibitor, to treat patients with BRAF-V600E mutations [82]. This was a significant study as it was the first prospective trial composed of patients with BRAF-mutant CCA [82]. The JAK/STAT pathway has also been studied, given that an aberrancy to this pathway was found in almost 50% of iCCA patients [83]. Numerous agents that inhibit the components of the JAK/STAT pathways have already been proposed for the treatment of various cancer types, and therefore, the treatment of CCA with these agents is also highly anticipated. Tropomyosin receptor kinase (TRK) inhibitors such as larotrectinib and entrectinib have already been approved for the treatment of solid tumors with NTRK gene fusions; however, we believe that additional research primarily consisting of CCA patients should be undertaken when specifically considering for CCA. 

Antibody drug conjugates (ADCs), with their ability to directly, specifically, and selectively deliver cytotoxic compounds to cancer cells, have also emerged as a novel therapy class [84]. This class of medication could almost be considered as a subtype of targeted therapy. In the current literature, there is limited mention of the role of ADCs specifically for the treatment of CCA, although in principle this is certainly possible. For example, there already exists an ADC capable of binding to HER2 to fight HER2-positive breast cancer [85]. Therefore, it makes sense for an antibody that targets HER2—such as the aforementioned zanidatamab recently approved under FDA breakthrough therapy designation for the treatment of advanced CCA—to be conjugated with a cytotoxic payload to increase the potency of the anticancer regimen. Further investigation exploring this class of medications for CCA is therefore required. 

Overall, numerous clinical trials investigating the efficacy of various targeted therapies are currently ongoing [78]. Some of these trials are investigating the efficacy of a particular targeted therapy when used as monotherapy, while others are being investigated in combination with other cancer regimens. Though some of the clinical trials are specifically evaluating its efficacy against CCA, most of them are also assessing other forms of cancer including pancreatic cancer, gallbladder cancer, lymphoma, and sarcoma. A recent article by Simile compiled a list of ongoing clinical trials of targeted therapies for the treatment of CCA [78]. The first targeted therapy that was FDA-approved for the treatment of CCA was on 17 April 2020, and within a 2-year time period, the number of FDA-approved targeted therapies for CCA has now grown to three: pemigatinib, infigratinib, and ivosidenib. During the same time period, two additional targeted therapies were granted breakthrough therapy designation: zanidatamab and futibatinib. This represents an explosion of growth in the availability of targeted therapies for CCA treatment in the US. Given the sheer number of clinical trials, more and more options for targeted therapies will be available. 

It is worth noting that no targeted therapies have yet been approved or granted a breakthrough therapy designation as a first-line treatment for cholangiocarcinoma, although there are ongoing clinical trials to directly compare them as first line against the standard of treatment. The FIGHT-302 trial (NCT03656536) is a global, multicenter, open-label, randomized phase III study that will compare the treatment of CCA with FGFR2 rearrangements with first-line pemigatinib compared to gemcitabine + cisplatin [86]. The clinical trial will enroll patients with unresectable CCA with documented FGFR2 fusions or other rearrangements and no prior systemic therapy within 6 months before enrollment (86). The study target is 432 patients and patients will be randomized into 2 groups: pemigatinib 13.5 mg daily on a 21-day cycle vs. IV gemcitabine 1000 mg/m^2^ + cisplatin 25 mg/m^2^ on day 1 and day 8 of 21-day cycles [86]. The primary endpoint from this study will be PFS, and its secondary endpoints include ORR, OS, duration of response, DCR, quality of life, and finally safety [86]. The PROOF-301 trial, a multicenter, open-label, randomized phase III study will directly compare infigratinib against the standard of care of gemcitabine and cisplatin as a first-line treatment [87]. The PROOF-301 trial (NCT03773302) will aim for a target study size of approximately 300 patients with unresectable or metastatic CCA with FGFR2 fusion or other rearrangements [68]. The patients will be randomized 2:1, with the experimental group taking oral infigratinib 125 mg daily for 21 days on 28-day cycles while the active comparator group takes IV gemcitabine 1000 mg/m^2^ + cisplatin 25 mg/m^2^ on day 1 and day 8 of 21-day cycles [68]. The primary endpoints from this study will be PFS, with secondary outcome measures consisting of the ORR, OS, duration of response, DCR, number of treatment-related AEs, and quality of life [68]. 

Of note, there has been increasing interest in the utilization of ctDNA—circulating tumor deoxyribonucleic acid—to help guide cancer therapy. These DNA molecules are specifically produced by tumor cells and can therefore allow clinicians to determine the type of mutation present in cancer cells, assist in clinical decision making, and help track disease state [88]. Given the nature of targeted therapies, the incorporation of ctDNAs could potentially have a tremendous impact on the utilization of targeted therapies at a larger scale. 

Among the largest challenges of incorporating targeted therapies, as observed for many other new medications, are the cost and insurance approval. Another potential challenge is potential long-term side effects. Some of the immediate side-effects, such as hyperphosphatemia, have been well documented; however, there are currently not enough data detailing the longer effects of these medications. Although targeted therapies are designed to specifically target cancer cells, there is no guarantee that it will completely bypass all normal cells. It is certainly possible that some clinicians would therefore be more apprehensive of utilizing these newer medications. Finally, another challenge is the resistance of cancer cells against targeted therapy. As with all antineoplastic therapies, the presence of resistance within certain patients is inevitable given the nature of the disease. 

Overall, the results from both the FIGHT-302 trial and PROOF-301 trial are greatly anticipated, as these will provide a direct first-line comparison between a single targeted therapy vs. standard chemotherapy. Positive results from either of these trials could potentially allow physicians to bypass the harmful cytotoxic effects of standard chemotherapy and instead pursue the “cleaner” targeted therapy as a first-line agent for CCA. The fight against cholangiocarcinoma has often felt like a losing battle, but as these newer treatments are introduced and evaluated, the overall outlook is slowly becoming more and more optimistic.

## 6. Conclusions

The FDA approval of targeted therapies for the treatment of CCA marks an exciting time for clinicians across the nation. New options are now becoming available to fight against this terrible disease, and it is only a matter of time until more targeted therapies are approved. Numerous clinical trials are being actively performed investigating the efficacy of various targeted therapies against CCA, with some trials seeking to directly compare targeted therapy against standard first-line therapy. The fight against cholangiocarcinoma has often felt like a losing battle, but as these newer treatments are introduced and evaluated, the overall outlook is slowly becoming more and more optimistic.

## Figures and Tables

**Figure 1 cancers-14-02641-f001:**
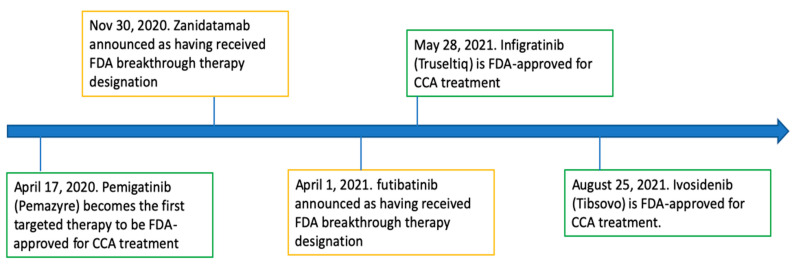
Timeline: targeted therapy with FDA approval or breakthrough therapy designation.

**Figure 2 cancers-14-02641-f002:**
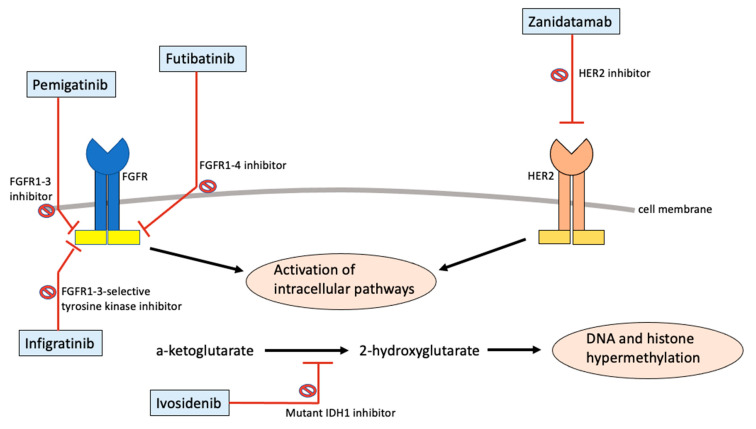
Mechanism of action of targeted therapies with FDA approval or breakthrough therapy designation.

**Table 1 cancers-14-02641-t001:** Relevant clinical studies for targeted therapies with FDA approval or breakthrough therapy designation for CCA.

Drug	Drug Class	Study	Treatment Group	ORR	DCR	PFS	OS
Pemigatinib	FGFR inhibitor	NCT03656536	107 participants	35.50%	82%	6.9 months	21.1 months
Infigratinib	FGFR inhibitor	NCT02150967	61 participants	14.80%	75.40%	5.8 months	Not reported
Ivosidenib	IDH1 inhibitor	NCT02989857	124 participants	2%	53%	2.7 months	10.3 months
Zanidatamab	HER2-targeted antibody	NCT04466891	20 participants	47%	65%	Not reported	Not reported
Futibatinib	FGFR inhibitor	NCT02052778	67 participants	34.30%	76.10%	Not reported	Not reported

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
