# Peer review of "Timeline of FDA-Approved Targeted Therapy for Cholangiocarcinoma"

_cancers, 2022, doi:10.3390/cancers14112641_

Round 1

Reviewer 1 Report

I read with great interest this original article “Timeline of FDA-Approved Targeted Therapy for Cholangiocarcinoma”. This is a well-written article, that focus on an important point since approved treatments are changing a lot in CCA, but some points are missing.

In addition, I have additional concerns, which need also to be discussed. Please consider the following comments.

Major comments

  1. Citing ‘immunotherapy” as first example for precision medicine is not very accurate to my point of view since it can be untargeted. Targeted therapies would be more accurate to cite first.
  2. Parts should be organized not by drugs but by molecular alteration (e.g FGFR, IDH etc) to ease reading.
  3. Are there any data on prescriptions volume after FDA approval for each drugs ?
  4. Please specific for pemigatinib and infigratinib and ivosidenib that patients in phase 2 or 3 trials were previously treated and median line of .treatment.
  5. Line 168 169 : give pvalue as line 175 176.
  6. NRTK inhibitors are pan-cancers approved drugs and should be also cited, since applicable to CCA in case of NTRK fusion.
  7. I would rather discuss mechanisms of action of targeted therapies before discussing their FDA approval.
  8. A table recapitulating OS, PFS, ORR, DCR etc and main characteristics of study population for each studies that lead to drug approval would be valuable.
  9. the introduction, authors should explicit systemic chemotherapies and main results for 1st and second line (OS, PFS, ORR etc) since treatments were approved in line to those results.

Author Response

Journal: Cancers (ISSN 2072-6694)

Manuscript ID: cancers-1710141

Type: Review

Title: Timeline of FDA-Approved Targeted Therapy for Cholangiocarcinoma

Section: Cancer Immunology and Immunotherapy

Special Issue: Targeted Treatment and Immunotherapy for Hepato-Biliary Tumors

Dear Reviewer 1,

We would like to thank the learned reviewer for consideration of our manuscript for publication and thoroughly appreciate the time taken to provide us with valuable comments to improve the readability of our contribution to literature. We have diligently provided point wise responses to all comments below

Response to Reviewer 1 Comments:

I read with great interest this original article “Timeline of FDA-Approved Targeted Therapy for Cholangiocarcinoma”. This is a well-written article, that focus on an important point since approved treatments are changing a lot in CCA, but some points are missing.

In addition, I have additional concerns, which need also to be discussed. Please consider the following comments.

Major comments:

  1. Citing ‘immunotherapy” as first example for precision medicine is not very accurate to my point of view since it can be untargeted. Targeted therapies would be more accurate to cite first.

Author’s Response: Thanks a lot for this very important point, Immunotherapy is not specifically targeted, however it is able to impact the cancer cells while sparing the healthy cells which is why we’ve utilized the term precision medicine. This is in contrast to the other modes of treatments that do affect the surrounding cells as we’ve elaborated previously. We included immunotherapy first because it was not the main focus of the paper, therefore we felt that briefly mentioning immunotherapy first before delving deeply into targeted therapy felt more appropriate.

  1. Parts should be organized not by drugs but by molecular alteration (e.g FGFR, IDH etc) to ease reading.

Author’s Response: Thank you very much for this comment,  Normally we would agree with organizing by the class of medication or the molecular alteration, however for this paper we have decided to organize by the drugs to showcase the chronological order in which the medications were approved by the FDA. Also, we couldn’t combine both FGFR inhibitors under one section since one medication did receive FDA approval while the other only obtained FDA breakthrough therapy designation.

  1. Are there any data on prescriptions volume after FDA approval for each drugs?

Author’s Response: Thanks a lot for this very important point, currently there isn’t data of prescription volume after FDA approval.

  1. Please specific for pemigatinib and infigratinib and ivosidenib that patients in phase 2 or 3 trials were previously treated and median line of. treatment.

Author’s response: Thanks a lot for this very important point, we made the change, now including the fact that the patient population had previously undergone treatment. The articles did not mention the median line of treatment, however, did include the percentage showing the number of prior systemic therapies of the participants. We’ve added this information to the manuscript.

  1. Line 168 169 : give pvalue as line 175 176.

Author’s Response: Thank you very much for this comment, for much of the data the p-value was not reported. Most of the data was shown with 95% confidence intervals, therefore we’ve made these additions to the manuscript.

  1. NRTK inhibitors are pan-cancers approved drugs and should be also cited, since applicable to CCA in case of NTRK fusion.

Author’s Response: Thanks a lot for this comment, We have added the NTRK inhibitors in the discussion, however, did not elaborate extensively as with the other drugs in the manuscript. The main rationale for this was that the approved NRTK inhibitors were not as specific to CCA, since the clinical trials that led to the FDA approval consisted of a minimal number of CCA patients.

  1. I would rather discuss mechanisms of action of targeted therapies before discussing their FDA approval.

Author’s Response: Thanks a lot for this comment,  Ideally we would have included the mechanism of action before discussion of the FDA approval, however given the limited number of FDA-approved medications we felt it more appropriate to discuss the medications first before delving into the mechanism of actions. This way we could include the other potential targets/mechanism of actions outside the current approved medications.

  1. A table recapitulating OS, PFS, ORR, DCR etc and main characteristics of study population for each studies that lead to drug approval would be valuable.

Author’s Response: Thanks a lot for this comment, A brief table has been added.

  1. the introduction, authors should explicit systemic chemotherapies and main results for 1st and second line (OS, PFS, ORR etc) since treatments were approved in line to those results.

Author’s Response: Thank you very much for this comment, we’ve added a description of the 1st and 2nd line treatments.

Thank you,

The team

Reviewer 2 Report

The manuscript by Cho et al is a review about FDA approved drugs for treatment of cholangiocarcinoma. In principle manuscript is scientifically sound, yet I would recommend some changes/suggestions to improve the manuscript:

  1. The mechanism of action of CCA FDA approved drugs should be largely extended. Since the review is rather short and there are only a few FDA approved drugs for CCA they should be described in much more details: how were they developed, what is exactly their mechanism of action (maybe some structures of targets with inhibitors should be shown if applicable), what are their benefits and drawbacks? Do CCA cells develop resistance against these compounds?
  2. In the Outlook section the authors could study and if applicable include information about cytotoxic conjugates (like ADC) for treatment of CCA, or, if there are none in literature or under clinical trials, the authors could discuss if FDA approved conjugates (like ADCs) targeting cancer markers like FGFRs or HER2 for treatment of other cancers (e.g. breast cancer) could be used for treatment of CCA.
  3. The Figure should be re-organized/revised. Since all 3 mentioned FDA approved inhibitors are either ATP-mimicks, or irreversible inhibitor binding P-look of the intracellular kinase domain, current representation is misleading, as it suggests that pemigatinib and futibatinib bind the extracellular domain of FGFRs, which is obviously not correct. This is in contrast to HER2 blocking Ab, which acts on the extracellular domain of HER2.

Author Response

Journal: Cancers (ISSN 2072-6694)

Manuscript ID: cancers-1710141

Type: Review

Title: Timeline of FDA-Approved Targeted Therapy for Cholangiocarcinoma

Section: Cancer Immunology and Immunotherapy

Special Issue: Targeted Treatment and Immunotherapy for Hepato-Biliary Tumors

Dear Reviewer 2,

We would like to thank the learned reviewer for consideration of our manuscript for publication and thoroughly appreciate the time taken to provide us with valuable comments to improve the readability of our contribution to literature. We have diligently provided point wise responses to all comments below

Response to Reviewer 2 Comments:

The manuscript by Cho et al is a review about FDA approved drugs for treatment of cholangiocarcinoma. In principle manuscript is scientifically sound, yet I would recommend some changes/suggestions to improve the manuscript:

Major comments:

  1. The mechanism of action of CCA FDA approved drugs should be largely extended. Since the review is rather short and there are only a few FDA approved drugs for CCA they should be described in much more details: how were they developed, what is exactly their mechanism of action (maybe some structures of targets with inhibitors should be shown if applicable), what are their benefits and drawbacks? Do CCA cells develop resistance against these compounds?

Author’s response: We really appreciate this comment, your last point regarding resistance is worth noting. As of right now, there isn’t much mention of resistance to these new agents, however, we do believe that targeted therapy resistance will occur in CCAs. There are already such cases occurring in other cancers such as breast cancer which has had long experience with targeted therapy. We will incorporate this point into our manuscript, specifically in the latter part of the manuscript details the potential barriers to the incorporation of targeted therapies. As for the mechanism of action, we wanted to keep it fairly broad for our manuscript since most clinicians in practice typically associate with a broader categorization.

  1. In the Outlook section the authors could study and if applicable include information about cytotoxic conjugates (like ADC) for treatment of CCA, or, if there are none in literature or under clinical trials, the authors could discuss if FDA approved conjugates (like ADCs) targeting cancer markers like FGFRs or HER2 for treatment of other cancers (e.g. breast cancer) could be used for treatment of CCA.

Author’s Response: Thank you very much for this comment, We agree that ADCs should be mentioned in the manuscript. There’s isn’t much literature currently exploring the utility of ADC for CCAs specifically, but in principle ADCs can certainly be applied for CCAs. We included a brief description of the ADCs and the possible utility of these for treating CCAs.  

  1. The Figure should be re-organized/revised. Since all 3 mentioned FDA approved inhibitors are either ATP-mimicks, or irreversible inhibitor binding P-look of the intracellular kinase domain, current representation is misleading, as it suggests that pemigatinib and futibatinib bind the extracellular domain of FGFRs, which is obviously not correct. This is in contrast to HER2 blocking Ab, which acts on the extracellular domain of HER2.

Author’s Response: Thank you very much for this comment, the appropriate corrections were made to the figure.

Thank you,

The team

Reviewer 3 Report

This is a well-written overview of targeted therapy of cholangiocarcinoma with special emphasis on FDA-approved targeted therapies. The organization of the article is appropriate. The figure is well designed and easy to understand. 

I have the following suggestions to enhance the review-

  1. Authors should include a brief discussion of all major targeted therapies for CCA, including BRAF  and NTRK-directed therapies.
  2. Immunotherapy targeting dMMR/MSI-H tumors should be briefly discussed.
  3. A table should be included summarizing ongoing clinical trials with targeted therapy combinations in the 'future steps' section.
  4. Authors should also discuss the barriers to implementing targeted therapies.
  5. A discussion of ctDNA should be brought in because ctDNA will likely help implement targeted therapies on a larger scale. 

Overall, a comprehensive review, and I support publishing this article.

Author Response

Journal: Cancers (ISSN 2072-6694)

Manuscript ID: cancers-1710141

Type: Review

Title: Timeline of FDA-Approved Targeted Therapy for Cholangiocarcinoma

Section: Cancer Immunology and Immunotherapy

Special Issue: Targeted Treatment and Immunotherapy for Hepato-Biliary Tumors

Dear Reviewer 3,

We would like to thank the learned reviewer for consideration of our manuscript for publication and thoroughly appreciate the time taken to provide us with valuable comments to improve the readability of our contribution to literature. We have diligently provided point wise responses to all comments below

Response to Reviewer 3 Comments:

This is a well-written overview of targeted therapy of cholangiocarcinoma with special emphasis on FDA-approved targeted therapies. The organization of the article is appropriate. The figure is well designed and easy to understand.

Major comments:

  1. Authors should include a brief discussion of all major targeted therapies for CCA, including BRAF and NTRK-directed therapies.

Author’s response: We really appreciate this comment, we included a brief description of BRAF targeted therapy that has recently been studied. There was a study performed by MD Anderson that utilized dabrafenib, a BRAF inhibitor, with trametinib, a MEK inhibitor, to treat patients with BRAF-V600E mutations. This was the first study that has worked with BRAF-mutant CCA, and we will describe this briefly in our future steps section.

As for the NTRK-directed therapies, these have already received FDA approval for the treatment of solid tumors, which would include cholangiocarcinoma. However, we’ve only delved into therapies that were approved specifically for CCA, since the relevant trials had consisted primarily of CCA patients. This was not the case for the NTRK-directed therapies. We’ve added a brief description detailing this.

  1. Immunotherapy targeting dMMR/MSI-H tumors should be briefly discussed.

Author’s response: We are focusing on specifically targeted therapies and therefore have not included this in our paper.

  1. A table should be included summarizing ongoing clinical trials with targeted therapy combinations in the 'future steps' section.

Author’s Response: Thank you very much for this comment, during our research, we came across a paper that had already compiled a list of ongoing clinical trials. We’ve mentioned the author as well as the citation for the table.

  1. Authors should also discuss the barriers to implement targeted therapies.

Author’s Response: Thank you very much for this comment, this was included in the manuscript. The barriers to implementation that we have included include costs/insurance, the uncertainty of long-term side effects, as well as therapy resistance.

  1. A discussion of ctDNA should be brought in because ctDNA will likely help implement targeted therapies on a larger scale.

Author’s Response: We really appreciate this comment, Given the nature of targeted therapies, ctDNA’s could potentially make a tremendous impact on the utilization of targeted therapies. This was included in the manuscript.

Thank you,

The team

Round 2

Reviewer 1 Report

Authors have responded to my comments, therefore I agree with publication.

Reviewer 3 Report

The authors have addressed all my critiques appropriately. This article is ready for publication.